# Critical role of CD4+ T cells and IFNγ signaling in antibody-mediated resistance to Zika virus infection

Carolina G.O. Lucas[1], Jamil Z. Kitoko[1,2], Fabricio M. Ferreira[1], Vinicius G. Suzart[1], Michelle P. Papa[3], Sharton V.A. Coelho[3], Cecilia B. Cavazzoni[4], Heitor A. Paula-Neto[5], Priscilla C. Olsen[2], Akiko Iwasaki [6], Renata M. Pereira[7], Pedro M. Pimentel-Coelho[8], Andre M. Vale[4], Luciana B. de Arruda[3] & Marcelo T. Bozza[1]

Protective adaptive immunity to Zika virus (ZIKV) has been mainly attributed to cytotoxic CD8+ T cells and neutralizing antibodies, while the participation of CD4+ T cells in resistance has remained largely uncharacterized. Here, we show a neutralizing antibody response, dependent on CD4+ T cells and IFNγ signaling, which we detected during the first week of infection and is associated with reduced viral load in the brain, prevention of rapid disease onset and survival. We demonstrate participation of these components in the resistance to ZIKV during primary infection and in murine adoptive transfer models of heterologous ZIKV infection in a background of IFNR deficiency. The protective effect of adoptively transferred CD4+ T cells requires IFNγ signaling, CD8+ T cells and B lymphocytes in recipient mice. Together, this indicates the importance of CD4+ T cell responses in future vaccine design for ZIKV.

[1] Laboratório de Inflamação e Imunidade, Departamento de Imunologia, Universidade Federal do Rio de Janeiro, Rio de Janeiro, RJ 21941-902, Brazil. [2] Laboratório de Bacteriologia e Imunologia Clínica, Faculdade de Farmácia, Universidade Federal do Rio de Janeiro, Rio de Janeiro, RJ 21941-902, Brazil. [3] Laboratório de Genética de Imunologia de Infecções Virais, Departamento de Virologia, Universidade Federal do Rio de Janeiro, Rio de Janeiro, RJ 21941-902, Brazil. [4] Laboratório de Sinalização e Imunoreceptores, Instituto de Biofísica Carlos Chagas Filho, Universidade Federal do Rio de Janeiro, Rio de Janeiro, RJ 21941-902, Brazil. [5] Laboratório de Alvos Moleculares, Departamento de Biotecnologia Farmacêutica, Faculdade de Farmácia, Universidade Federal do Rio de Janeiro, Rio de Janeiro, RJ 21941-902, Brazil. [6] Department of Immunobiology, Howard Hughes Medical Institute, Yale University School of Medicine, New Haven, CT 06519, USA. [7] Laboratório de Imunologia Molecular, Departamento de Imunologia, Universidade Federal do Rio de Janeiro, Rio de Janeiro, RJ 21941-902, Brazil. [8] Laboratório de Neurobiologia Celular e Molecular, Instituto de Biofísica Carlos Chagas Filho, Universidade Federal do Rio de Janeiro, Rio de Janeiro, RJ 21941-902, Brazil. These authors contributed equally: Carolina G. O. Lucas, Jamil Z. Kitoko. Correspondence and requests for materials should be addressed to M.T.B. (email: mbozza@micro.ufrj.br)

Zika virus (ZIKV) was first isolated 70 years ago in the Zika Forest of Uganda[1] and, until recently, was only occasionally isolated from human patients both in Africa and Asia. Recent ZIKV outbreaks in the Americas, however, affected millions of individuals in several countries, resulting in a considerable number of cases of Guillain–Barré Syndrome, and sporadic cases of meningoencephalitis and myelitis in infected adult patients[2–6]. Importantly, a dramatic raise in the number of congenital malformations, especially microcephaly, first reported in the northeast of Brazil, is associated with ZIKV infection during pregnancy[7–9]. These cases of congenital ZIKV syndrome result, at least in part, from the ability of ZIKV to infect and trigger cell death of neuronal cell progenitors during development[10,11]. The broad tissue tropism, the long-term persistence in a number of different cells and fluids, including brain, lymph nodes, testis and semen, and the sexual transmission, places ZIKV as a unique virus among flaviviruses and a serious public health concern[12–16].

Type 1 IFN response is associated with an innate resistance essential for infection control. Susceptibility of humans to ZIKV is in part due to the effect of ZIKV NS5 protein in increasing proteasome-mediated degradation of STAT2, a transcription factor essential to type 1 IFN receptor signaling[17,18]. Mouse STAT2 is not a target for ZIKV NS5 and thus immune competent mice are highly resistant to ZIKV infection. Therefore, murine models of ZIKV infection generally relying on the use of mouse strains deficient of type 1 IFN signaling[19–22]. This limitation can be circumvented by inoculation of extremely high titers of ZIKV, infection of neonatal mice or intracerebral virus inoculation, all of which have been reported to cause infection and disease in immune competent mouse strains[19,21,23].

In the past 2 years, understanding on the adaptive immune response to ZIKV infection has been obtained with experimental animal models and clinical studies, although important gaps in knowledge remain. $Rag1^{-/-}$ mice, deficient of T and B cells, are resistant to ZIKV infection, unless type 1 IFNR signaling is also blocked[24]. Similarly, in mice treated with anti-IFNAR1 antibody, lack of CD4+ T cells, CD8+ T cells, or B cells have no impact in viral loads upon a secondary intravaginal challenge with a homologous ZIKV, while the absence of both T and B cells renders mice highly susceptible to secondary ZIKV infection[25]. In a different model, however, the absence of CD8+ T cells in ZIKV-infected mice treated with IFNAR-blocking antibody increases viral loads and lethality, while adoptive transfer of central memory CD8+ T cells enhances viral clearance[26]. Various studies reported a role for neutralizing antibodies in heterologous immunization or in cross-protective infections. Previous ZIKV infection in humans and experimental animals or immunization generate neutralizing antibodies, especially against epitopes on the envelope protein dimer or in domain III (EDIII), which are efficient in preventing ZIKV infection and disease[27–31]. Heterologous protection of nonhuman primates infected with an African ZIKV strain against a challenge with a more severe Asian lineage has been demonstrated[32]. Moreover, cross-protective responses to ZIKV were observed by human antibodies to Dengue virus (DENV), although antibody-dependent enhancement was also described[30,31,33–36].

The current paradigm of the adaptive immune response to flavivirus infection is one in that cytotoxic CD8+ T cells and the antibody response are essential to early and long-term resistance[26,37–41]. The participation of CD4+ T cells in the protective response to flavivirus infection has been less uniform[42]. In experimental DENV infection, mice lacking CD4+ T cells have no increased susceptibility, and depletion of CD4+ T cells has no impact on viral control, neutralizing antibody production, germinal center formation or CD8+ T cell activation in $Ifnar1^{-/-}$

mice[39,43]. However, immunization with CD4+ T cellepitopes results in lower viral loads[39]. Adoptive transfer of immune CD4+ T cells is protective in a mouse model of intracerebral Japanese encephalitis virus infection only when transfer in combination with immune CD8+ T cells[44]. In a mouse model of West Nile Virus (WNV), it has been shown that CD4+ T cells contribute to the viral clearance from the central nervous system (CNS) through the improvement of antibody production and CD8+ T cell response[45]. However, CD4+ T cells can also control WNV viral loads and encephalitis independent of B and CD8+ T cells[46]. WNV-specific CD4+ T cells produce IFNγ and display direct cytotoxic activity in vitro and in vivo. Importantly, mice lacking Foxp3+CD4+ T regulatory cells have increase lethality induced by WNV infection[47]. The actual role of antiviral CD4+ T cells in conferring protective immunity against ZIKV requires further characterization. It is conceivable that in ZIKV infection the role of CD4+ T cells could range from nonessential as in DENV to multifaceted as in WNV.

In the present study, we characterized the adaptive immune response of type 1 IFNR mutant mouse strain, A129, to the infection with the Brazilian ZIKV strain PE243. Infection with ZIKV PE243 causes weight loss in 4-week-old mice, that in most cases recover and clear the infection, while is always lethal to 3-week-old mice. The survival of 4-week-old mice was associated with a robust activation of adaptive immunity. Mechanistically, B cell, CD4+, and CD8+ T cell responses and IFNγ signaling were fundamental to host protection against a primary acute infection with ZIKV PE243. Using an adoptive cell transfer model, we uncovered a critical role of immune CD4+ T cells and neutralizing antibodies on heterologous resistance to an otherwise lethal challenge with the mouse-adapted ZIKV strain MR766. Infection with this strain caused high cerebral viral loads, microhemorrhages, and edema. Our data suggest that CD4+ T cells from PE243-infected mice activate a robust IFNγ-dependent B cell response, associated with the production of neutralizing IgG2a antibodies.

## Results

**ZIKV PE243 evokes a robust immune response in A129 mice.** We infected type 1 IFNR-deficient mice (A129) with the ZIKV strain PE243, a strain isolated in 2015 from a patient in Brazil that had a mildfebrile disease, and shows high homology to the Asian strains[48]. Groups of 3- and 4-week-old mice were injected intravenously with $2 \times 10^5$ plaque-forming units (PFU) of PE243. Three-week-old infected mice presented clinical signs of the disease, including weight loss, shivering and paralysis, succumbing between 3 and 8 days postinfection, while 4-week-old mice presented weight loss, recovering around 7 days after infection (Fig. 1a, b). High levels of viral RNA were detected in the spleens, kidneys, livers of both groups, although viral RNA was significantly higher in the brains of 3-week-old infected mice (Fig. 1c).

The observed survival of 4-week-old A129 mice infected with PE243 suggested that the early adaptive immune response triggered by ZIKV was efficient in promoting resistance to this ZIKV strain. Thus, we analyzed the activation profile of lymphocyte subpopulations in the spleens at day 7 postinfection. The proportion of effector CD8+ T cells, based on CD62L/low and CD44/high expression, was significantly increased in mice infected with PE243 (Fig. 1d). Upon polyclonal stimulation ZIKV infection triggered an expansion of CD8+ T cells expressing IL-2, CD107a, Granzyme B and Perforin, but not IFNγ (Fig. 1e). The proportion of effector CD4+ T cells was also increased in ZIKV-infected mice (Fig. 1f). Moreover, a higher proportion of these CD4+ T cells from PE243-infected mice produced IFNγ and in a

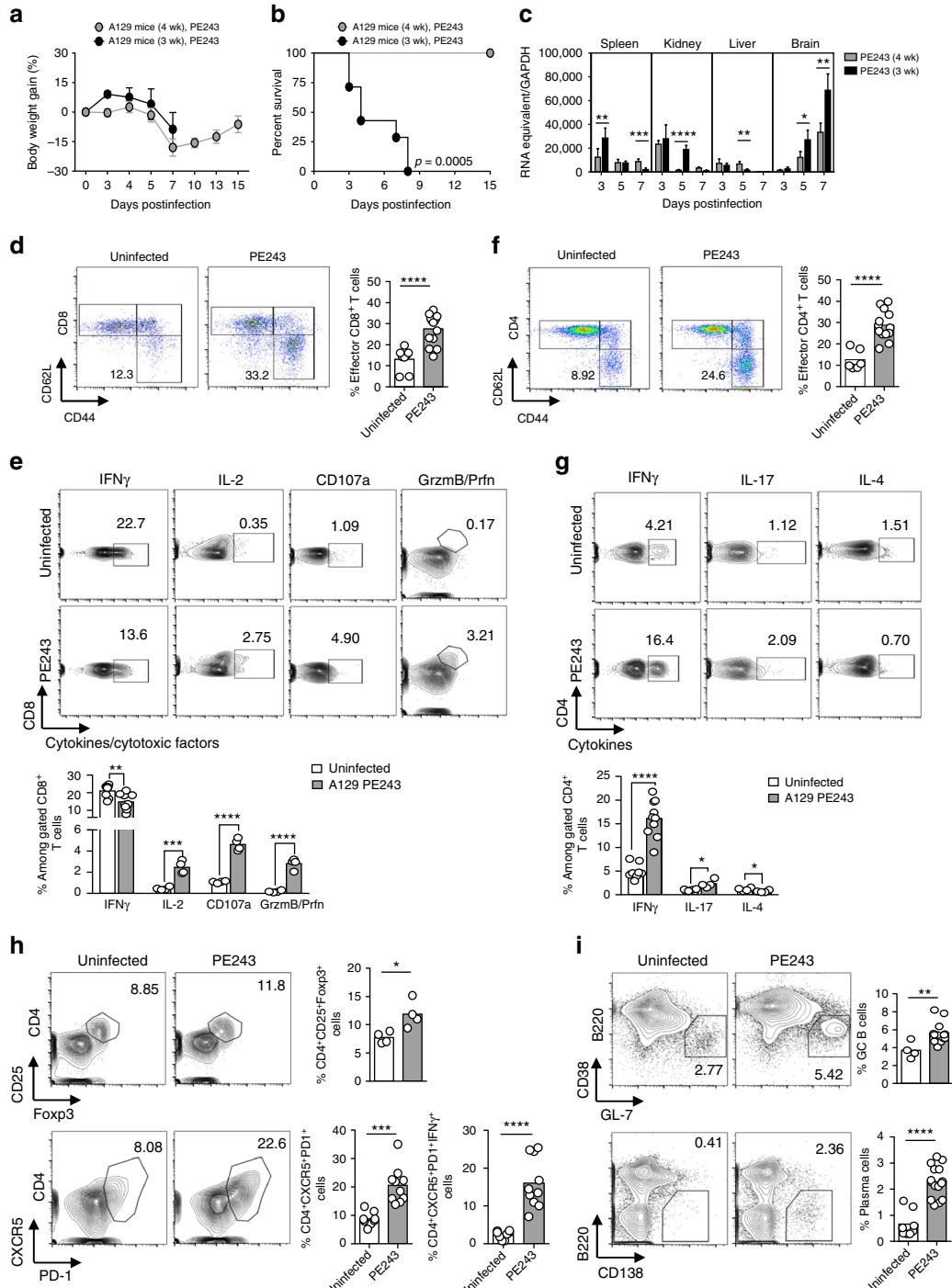

**Fig. 1** A129 mice 4-week-old survive to ZIKV PE243 and evoke a robust immune response. Three- and 4-week-old A129 mice were inoculated intravenously with $2 \times 10^5$ PFU of ZIKV strain PE243. Uninfected mice were used as control. **a, b** Body weight gain (**a**) and lethality (**b**) of 3- ($n = 9$) or 4-week-old A129 mice ($n = 15$) infected with ZIKV PE243 were monitored for up to 15 days postinfection. **c** ZIKV RNA copies in the spleen, kidney, liver, and brain determined by qRT-PCR at days 3, 5, and 7 postinfection of 3- ($n = 3–8$) or 4-week-old mice ($n = 4–6$). Results are expressed as RNA equivalent/copies and normalized by GAPDH. Flow cytometry were performed on splenocytes from uninfected ($n = 4–8$) or 4-week-old mice at day 7 postinfection ($n = 4–15$). For cytokines and cytotoxic factors detection, splenocytes were restimulated in vitro with PMA and ionomycin in the presence of brefeldin A during 4 h. **d** Representative dot plot and frequency of CD62L⁻CD44⁺ (effector) among CD8⁺. **e** Representative counter plot and frequency of IFNγ, IL-2, CD107a, granzyme B (GrzmB), and perforin (Prfn) among CD8⁺ T cells. **f** Representative dot plot and frequency of CD62L⁻CD44⁺ (effector) among CD4⁺. **g** Representative counter plot and frequency of I IFNγ, IL-4, and IL-17 among CD4⁺ T cells. **h** Representative counter plot and frequency of CD25⁺Foxp3⁺ or CXCR5⁺PD-1⁺ and CXCR5⁺PD-1⁺IFNγ⁺ among CD4⁺ T cells. **i** Representative counter plot and frequency of germinal center B cells among B220⁺CD138⁻ and plasma cells. Results are shown as mean in **d–i** or as mean ± standard deviation in **a** and **c**. Data are representative of two independent experiments in **a**, **b**, and **c**, data are presented as a pool of two or three independent experiments in **d–i**. Survival data were analyzed by log rank test. Data were analyzed by Student's $t$ test in c–i. * $p ≤ 0.05$; ** $p ≤ 0.01$; *** $p ≤ 0.001$; **** $p ≤ 0.0001$

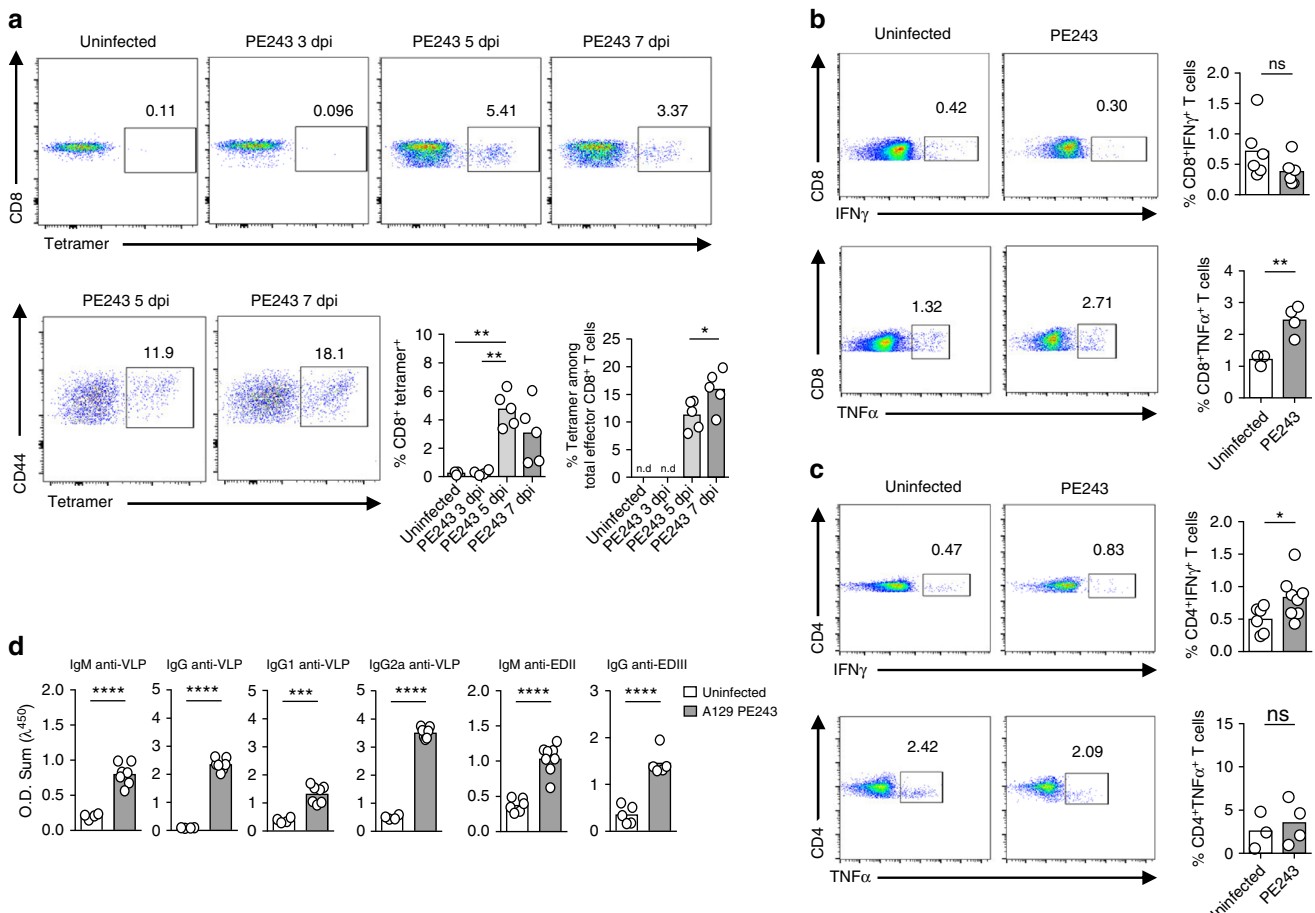

**Fig. 2** Four-week-old A129 mice elicit specific immune response against ZIKV PE243. Four-week-old A129 mice were inoculated intravenously with $2 \times 10^5$ PFU of ZIKV strain PE243. Uninfected mice were used as control. **a** Representative dot plot and frequency of protein E ZIKV tetramer positive cells among CD8$^+$ T cells or total CD8$^+$CD62L$^-$CD44$^+$ cells (effector). Data were obtained from ZIKV PE243-infected A129 mice at days 3 ($n = 4$), 5 ($n = 5$), and 7 ($n = 5$) postinfection and compared to uninfected mice ($n = 3$). For intracellular cytokines analyzes by means of flow cytometry, splenocytes were obtained from uninfected A129 mice ($n = 3–6$) or ZIKV PE243-infected A129 mice ($n = 4–8$) at day 7 postinfection, restimulated in vitro using UV-inactivated ZIKV during 72 h in the presence of brefeldin A added in the last 8 h. **b** Representative dot plot and frequency of IFNγ and TNFα among CD8$^+$ T cells. **c** Representative dot plot and frequency of IFNγ and TNFα among CD4$^+$ T cells. **d** Serum IgM and IgG anti-ZIKV VLP or anti-ZIKV-EDIII as well as serum IgG1 and IgG2a anti-ZIKV-VLPs in ZIKV PE243-infected A129 mice ($n = 8$), compared to uninfected A129 mice ($n = 4$). Results are shown as mean. Data are representative of two independent experiments in **d**, data are presented as a pool of two independent experiments in **b** and **c**, experiments were performed once in **a**. Data were analyzed by Student's t test. * $p \leq 0.05$; ** $p \leq 0.01$; *** $p \leq 0.001$; **** $p \leq 0.0001$; ns not significant

lesser extent IL-17, while infection caused a reduction of CD4$^+$ T cells producing IL-4 (Fig. 1g). We observed a significant expansion of T regulatory cells and T follicular cells (T$_{FH}$), CD4$^+$ T cells that express both CD25 and Foxp3 or CXCR5 and PD-1, respectively, in the spleen of ZIKV-infected mice infected (Fig. 1h). Moreover, approximately 15% of the CD4$^+$ T cell population in the spleen of ZIKV PE243-infected mice was IFNγ producing, CXCR5$^+$ and PD-1$^+$ cells, about three times higher than the levels observed in uninfected mice. ZIKV triggered a significant increase in germinal center B (GC-B) cells (B220$^+$CD38$^+$GL-7$^+$) and plasma cells (Fig. 1i). These results demonstrate an early T cell and B cell response upon infection of A129 mice with the ZIKV strain PE243.

**Characterization of an early anti-ZIKV-specific response.** Infection with the ZIKV strain PE243 promoted a significant increase in the number of virus-specific CD8$^+$ T cells, based on ZIKV protein E tetramer staining, as early as 5 days postinfection (Fig. 2a). At 7 days postinfection, the effector cells stained with the tetramer reached approximately 15%. Moreover, the in vitro stimulation of splenocytes obtained from mice after 7 days of

infection with ultraviolet (UV)-inactivated ZIKV demonstrated an expansion of CD8$^+$ T cells expressing TNF, when compared to the cells obtained from uninfected control mice (Fig. 2b). Viral stimulation did not increase CD8$^+$ T cells expressing IFNγ in splenocytes from infected mice (Fig. 2b). Conversely, we observed a significant expansion of virus-specific CD4$^+$ T cells expressing IFNγ, while the proportion of TNF-expressing cells were not increased in the ZIKV-infected mice (Fig. 2c). Next, we performed ELISAs using PE243 virus-like particles (VLPs) to characterize the antibody response. Serum obtained from A129 mice after 7 days of ZIKV infection had significant higher titers of total IgM and IgG anti-ZIKV compared to serum from uninfected controls (Fig. 2d). Both, ZIKV-specific IgG1 and IgG2a isotypes were significantly increased in the infected mice, although higher titers were observed in the IgG2a isotype. Several studies demonstrated that anti-EDIII antibodies effectively neutralize flavivirus infections, including ZIKV[27–31]. Sera from A129 mice infected with PE243 had high titers of anti-EDIII IgM and IgG antibodies (Fig. 2d). Our results indicate that ZIKV PE243 primary infection triggers a robust and early adaptive immune response in A129 mice.

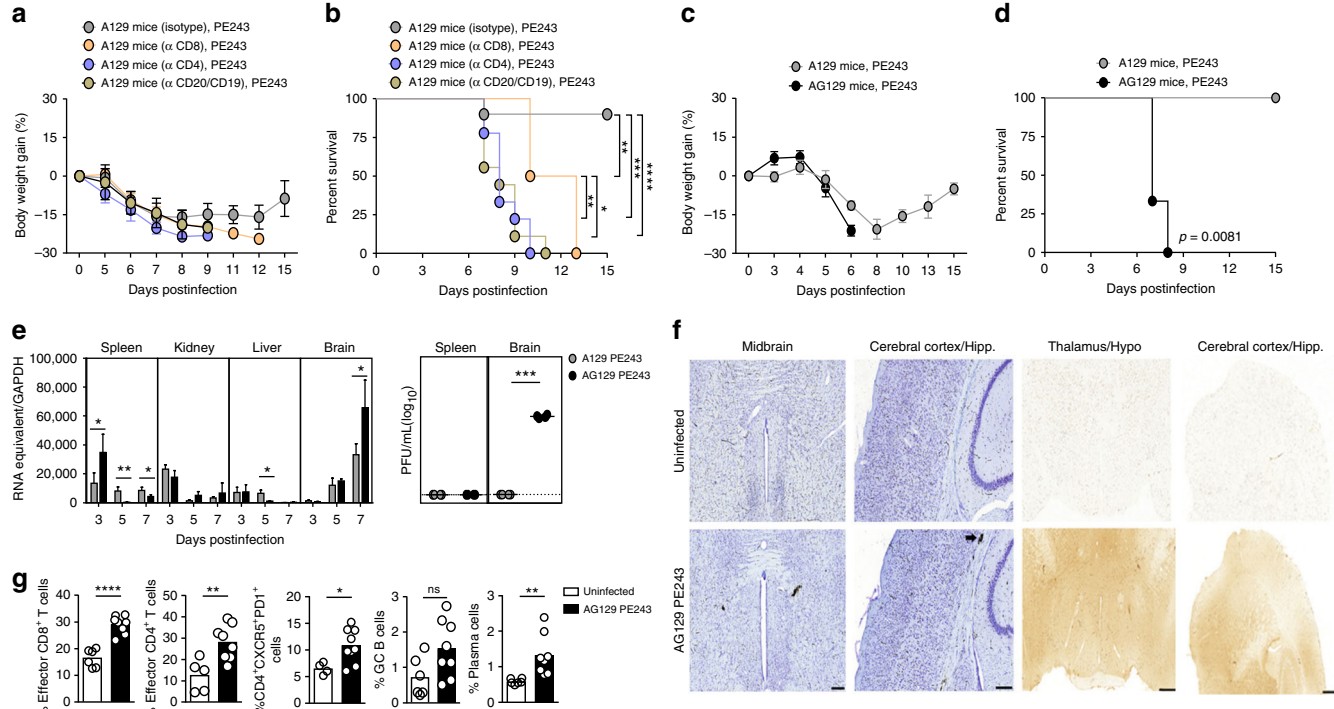

**Fig. 3** Lymphocytes and IFNγ signaling are required in the resistance to ZIKV PE243. Four-week-old A129 or AG129 mice were inoculated intravenously with $2 \times 10^5$ PFU of ZIKV PE243. **a** Body weight gain and **b** lethality of A129 mice previously depleted of CD8$^+$ ($n = 4$), CD4$^+$ T cells ($n = 9$), or B cells ($n = 9$) infected, and monitored for up to 15 days postinfection. Control mice received IgG2a isotype control ($n = 10$). **c** Body weight gain and **d** lethality of infected AG129 mice ($n = 11$) compared to A129 mice ($n = 10$). **e** ZIKV RNA copies in the spleen, kidney, liver, and brain postinfection of A129 ($n = 4–6$) or AG129 mice ($n = 3–4$), and infectious particles of ZIKV in the spleen and brain at 7 days postinfection, determined, respectively, by qRT-PCR and plaque assay. The horizontal dotted line indicates the limit of detection. **f** Representative photomicrographs of DAB-stained (Scale bar: 200 μm) or IgG-stained (Scale bar: 500 μm) brain sections retrieved from uninfected AG129 mice ($n = 3$) or AG129 mice at day 7 postinfection with ZIKV PE243 ($n = 3$). The arrow points to a focus of microhemorrhage in the white matter while IgG stains in brown. **g** Frequency of effector CD8$^+$ T cells or CD4$^+$ T cells, CD4$^+$CXCR5$^+$PD1$^+$ cells, germinal center B cells and plasma cells in the spleen of infected AG129 mice ($n = 8$) at day 7 postinfection, compared to uninfected AG129 mice ($n = 5–6$). Results are shown as mean in **g** and **e** or as mean ± standard deviation in **a**, **c**, and **e**. Data are presented as a pool of two (**a**, **b**, and **g**) or three (**c** and **d**) independent experiments, experiments were performed once in **e** and **f**. Survival data were analyzed by log rank test. Data were analyzed by Student's *t* test in **e** and **g**. (*) $p \leq 0.05$; (**) $p \leq 0.01$; (***) $p \leq 0.001$; (****) $p \leq 0.0001$; ns not significant; Hypo hypothalamus; hipp hippocampus

**Lymphocytes and IFNR2 are essential in primary ZIKV infection**. Next, the contribution of adaptive immune cell populations in the early protective response to primary ZIKV infection was analyzed. Selectively depletion of CD8$^+$ T, CD4$^+$ T, or B cells rendered 4-week-old A129 mice susceptible to the infection with PE243 (Fig. 3a, b). These results confirmed the protective role of CD8$^+$ T cells in ZIKV infection[26], and revealed the fundamental role of CD4$^+$ T cells and B cells in the primary response to an otherwise sublethal ZIKV infection. In fact, depletion of CD4$^+$ T cells or B cells caused significantly faster lethality when compared to the group of mice treated with anti-CD8 (Fig. 3b).

Contrasting with the high survival rates of 4-week-old A129 mice infected with ZIKV strain PE243, AG129 mice at this same age were highly susceptible to this ZIKV strain. AG129 mice, lacking type 1 and 2 IFNR, showed increased lethality, weight loss and viral RNAs, including high titers in the brains at 7 days postinfection (Fig. 3c–e). By plaque assay, PE243-infected AG129 mice presented high levels of infectious viruses particles, while virus remained barely detectable in the brains of A129 mice, suggesting the production of defective, noninfectious viral particles or very low levels of infectious viruses. Moreover, we observed increased microhemorrhagic lesions and blood–brain barrier (BBB) leakage in the AG129-infected group, compared to uninfected controls or to ZIKV PE243-infected A129 mice

(Fig. 3f, Supplementary Fig. 1d). Infection of AG129 mice with PE243 resulted in CD8$^+$ and CD4$^+$ T cell activation, including the activation of T$_{FH}$, but was unable to significantly increase the number of GC-B cells (Fig. 3g). Together, these results indicate that type 2 IFNR signaling contributes to the resistance against ZIKV infection and to the B cell response, including germinal center formation.

**CD4$^+$ T cells coordinate the resistance to heterologous ZIKV**. The ZIKV strain MR766 was derived from a virus isolated in 1947 in the Zika Forest of Uganda and was adapted to mice through more than 100 passages in suckling mice[1]. This strain is lethal to type 1 IFN deficient mice[19,22]. We confirmed that infection of 4-week-old A129 mice with MR766 caused a fast and severe disease, characterized by early weight loss, neurological signs and sudden death of all animals by 6 days postinfection (Supplementary Fig. 1a, b). Simultaneous infection with the PE243 strain was not effective in protecting against the fast lethality caused by the MR766 strain (Supplementary Fig. 1a, b). Viral RNAs were detected in the spleens, guts and brains of infected mice and the plaque assay consistently revealed high infectious particles in the brains of MR766-infected mice, reaching $10^6$ PFU/mL at 5 days postinfection (Supplementary Fig. 1c). Histopathological analyzes demonstrated the presence of BBB leakage and

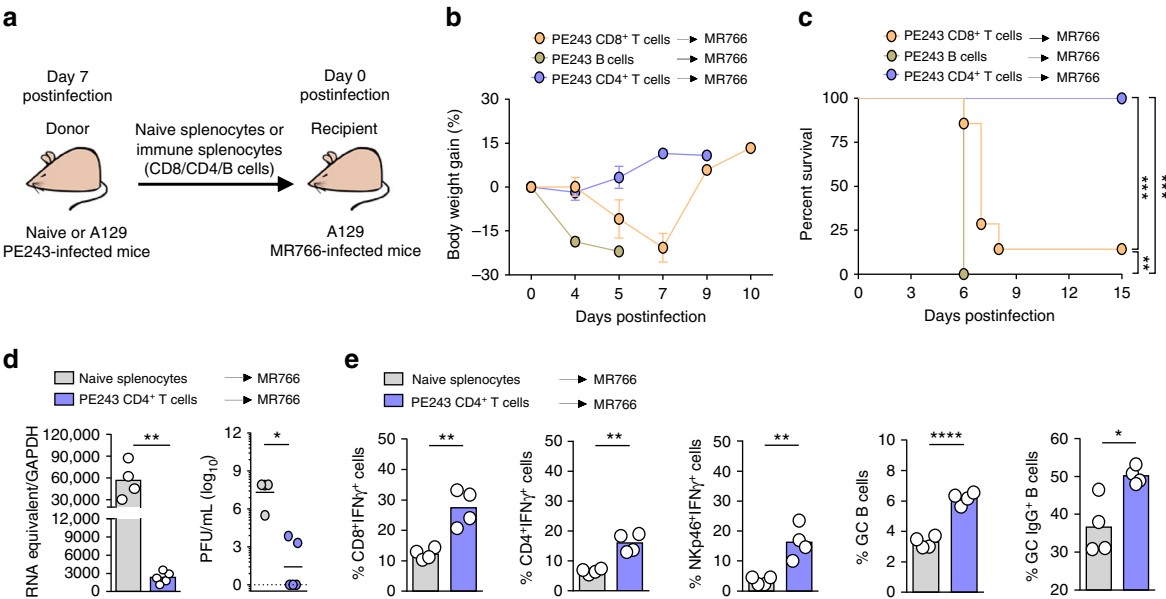

**Fig. 4** CD4+ T cell plays an essential role in the resistance against ZIKV MR766 infection. Recipient A129 mice were inoculated intravenously with $2 \times 10^5$ PFU of lethal ZIKV strain MR766 simultaneously to the receipt of CD8+ T cells, B cells or CD4+ T cells obtained from ZIKV strain PE243-infected A129 donor mice at 7 days postinfection. Uninfected mice were used as donor of naive splenocytes. **a** Schematic representation of adoptive transfer protocol. **b** Body weight gain and **c** lethality of recipient A129 mice that received PE243-immune CD8+ T cells ($1 \times 10^7$/mouse) ($n = 7$), CD4+ T cells ($4-5 \times 10^7$/mouse) ($n = 9$) or B cells ($1 \times 10^7$/mouse) ($n = 4$), monitored for up to 15 days postinfection. **d** ZIKV RNA copies and presence of ZIKV infectious particles in the brain of recipient mice that receive PE243-immune CD4+ T cells ($n = 5$) or naive splenocytes ($n = 4$) at 5 days postinfection determined by qRT-PCR and plaque assay, respectively. Results are expressed as RNA equivalent/copies and normalized by GAPDH and as PFU/mL for qRT-PCR and plaque assay, respectively. The horizontal dotted line indicates the limit of detection. **e** Spleens were retrieved from recipient A129 mice that received naive splenocytes ($n = 4$) or PE243-immune CD4+ T cells ($n = 4$) at 5 days postinfection for flow cytometry's assays. Relative frequency of IFNγ producing CD8+ T cells, CD4+ T cells and NKp46+ as well as germinal center B cells and IgG producing B cells. Results are shown as mean in **d** and **e** or as mean ± standard deviation in **b**. Data are presented as a pool of two independent experiments in **b** and **c**, experiment were performed once in **d** and **e**. Survival data were analyzed by Log rank test. Data were analyzed by Student's t-test in **d** and **e**. * $p \leq 0.05$; ** $p \leq 0.01$; *** $p \leq 0.001$; **** $p \leq 0.0001$

microhemorrhagic lesions in the CNS at 5 days of infection, especially in white matter tracts and in the brain stem, which were less prevalent in PE243-infected mice and never found in uninfected controls (Supplementary Fig. 1d).

To dissect the participation of T and B cells in the protective response to ZIKV, we decided to use a model of adoptive transfer followed by heterologous challenge with the lethal strain MR766. We performed adoptive transfers of total splenocytes or purified T or B cells from ZIKV PE243-infected A129 mice. Seven days postinfection cells from donor mice were transfer to A129 naïve recipient mice that were simultaneously challenged with the lethal ZIKV MR766 strain (Fig. 4a). Transfer of total splenocytes from ZIKV PE243-infected mice, but not from naïve controls, was very effective in preventing weight loss and lethality upon ZIKV MR766 infection (Supplementary Fig. 2a, b). Adoptive transfer of CFSE-labeled splenocytes demonstrated strong CD4+ T cell proliferation after 3 days of challenge with MR766, CD8+ T cells and B cells also proliferate (Supplementary Fig. 2c). Transferred CD4+ T cells readily produced and sustained IFNγ production in recipient mice, while CD8+ T cells only produced large amounts of IFNγ in the fourth generation observed by the CFSE profile of transferred splenocytes (Supplementary Fig. 2d).

Adoptive transfer of purified CD4+ T cells, but not B cells, prevented infection-associated weight loss and protected all animals against a lethal infection with ZIKV MR766 (Fig. 4b, c). Purified CD8+ T cells conferred only 10% protection to recipient mice (Fig. 4b, c). The number of cells transferred was based in the absolute numbers for each cell population found in the spleens of infected mice. Since this adoptive transfer protocol might be carrying viral particles, we characterized their presence in the

whole spleen and purified cell populations, revealing ZIKV RNAs by PCR in splenocytes (Fig. 1c), B cells (557 ZIKV RNA + copies/ $10^6$ cells) and CD4+ T cells (258 ZIKV RNA + copies/$10^6$ cells), but not in CD8+ T cells. As we showed, the simultaneous infection with PE243 and MR766 caused a similar lethality as MR766 alone, further excluding a putative early immune response against PE243 carried in the cell transfer as the cause of protection against MR766 (Supplementary Fig. 1a,b). The analyzes of viral titers in the brains demonstrated that mice receiving CD4+ T cells from PE243-infected mice had significant lower titers 5 days post transfer compared to mice that received cells from naïve mice (Fig. 4d). The immune response of recipient A129 mice after 5 days of CD4+ T cell adoptive transfer and challenge with ZIKV MR766 was characterized by increased percentage of spleen CD8+ T, CD4+ T and NK cells capable of producing IFNγ, GC-B cells and plasma cells (Fig. 4e). Given the protective effect of CD4+ T cells in the primary infection and in the adoptive transfer protocol, we reasoned that these cells coordinate the protective immune response against ZIKV.

**Requirements for CD4+ T cell-induced protection**. To further dissect the role of IFNγ signaling in the immune response to ZIKV, AG129 mice was used as recipients in the adoptive cell transfer protocol. Transfer of splenocytes or purified CD4+ T cells from A129 mice infected with ZIKV PE243 did not prevent weight loss or lethality of AG129 mice challenged with ZIKV MR766 (Fig. 5a, b). In this context, CD8+ T cells and B cells are the main candidates to the effect of IFNγ signaling in recipient mice. Depletion of CD8+ T cells with anti-CD8 antibody or B

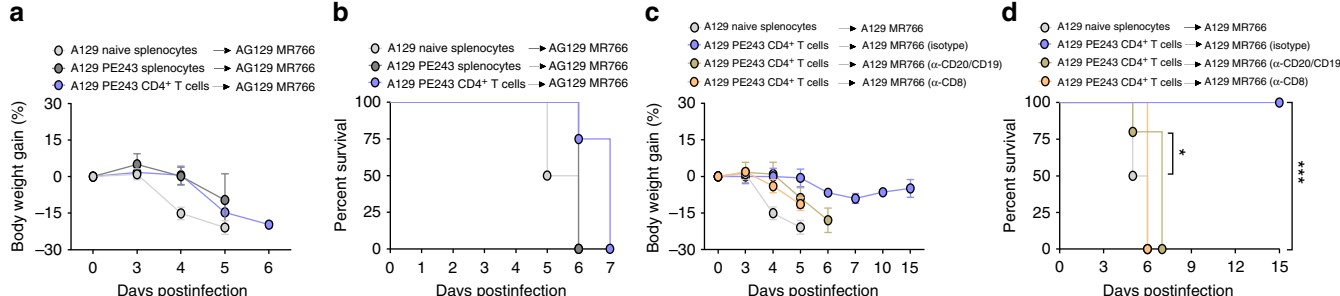

**Fig. 5** IFNγ, CD8⁺ T and B cell are crucial in CD4⁺ T cell protection against ZIKV MR766. Recipient AG129 mice or A129 mice depleted of B cell or CD8⁺ T cells were inoculated intravenously with $2 \times 10^5$ PFU of lethal ZIKV MR766. Simultaneously, the animals received immune splenocytes ($5 \times 10^7$/mouse) or immune CD4⁺ T cells ($5 \times 10^7$/mouse) obtained from ZIKV PE243-infected A129 donor mice at 7 days postinfection. Uninfected mice were used as donor of naive splenocytes to MR766-infected A129 control mice ($n = 8$). **a** Body weight gain and **b** lethality of recipient AG129 mice that received immune splenocytes ($n = 6$) or immune CD4⁺ T cells ($n = 4$) monitored for up to 7 days postinfection. **c** Body weight gain and **d** lethality of recipient A129 mice depleted of B cells ($n = 5$) or CD8⁺ T cells ($n = 4$) that received immune CD4⁺ T cells monitored for up to 15 days. IgG2a isotype control were used to exclude non-specific effects of B cell depleting antibodies ($n = 8$). Results are shown as mean ± standard deviation in **a** and **c**. Data are presented as a pool of two independent experiments in **a**–**d**. Survival data were analyzed by log rank test. * $p \leq 0.05$; *** $p \leq 0.001$

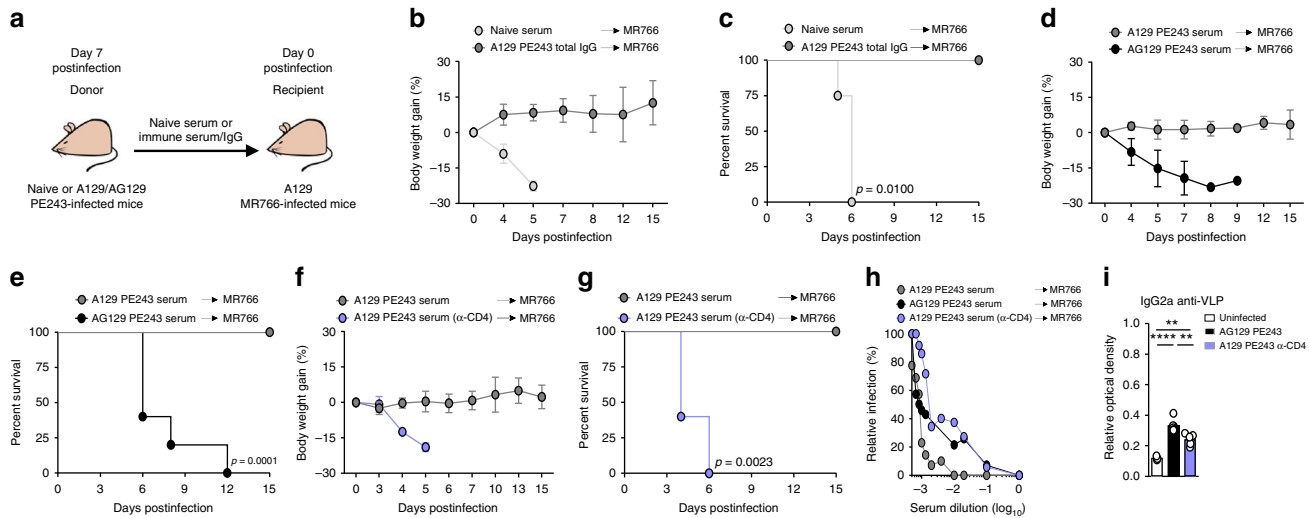

**Fig. 6** Serum of ZIKV PE243-infected A129 mice protects against ZIKV MR766 infection. **a** Schematic representation of passive serum administration. **b** Body weight gain and **c** lethality of recipient mice that received PE243 purified IgG (35.7 mg/kg) ($n = 4$) or naive serum ($n = 4$) were monitored for up to 15 days postinfection. **d** Body weight gain and **e** lethality of recipient mice that received serum from PE243-infected A129 mice ($n = 8$) or AG129 mice ($n = 5$) were monitored for up to 15 days postinfection. **f** Body weight gain and **g** lethality of recipient mice that received serum from PE243-infected A129 mice previously depleted of CD4⁺ T cells ($n = 5$) or not ($n = 5$) were monitored for up to 15 days postinfection. **h** Pooled sera from A129 or AG129 mice infected with ZIKV PE243 as well as infected A129 mice previously depleted of CD4⁺ T cells were collected, serially diluted and incubated with ZIKV MR766. Infectivity was determined by plaque assay. **i** Serum IgG2a anti-ZIKV VLPs in uninfected mice ($n = 3$), ZIKV strain PE243-infected AG129 mice ($n = 6$) and ZIKV PE243-infected A129 mice previously depleted of CD4⁺ T cells ($n = 5$). Results are represented as relative optical density by performing the ratio with ZIKV strain PE243-infected A129 mean value $\left( \frac{x\ group\ O.D.\ sum\ value}{A129\ PE243\ O.D.\ sum\ mean\ value} \right)$. Results are shown only as mean in **i** or as mean ± standard deviation in **b**, **d**, and **f**. Data are presented as a pool of two independent experiments in **b**–**g** or as representative of two independent experiments in **h** and **i**. Survival data were analyzed by log rank test. Data were analyzed by Student's t test in **i**. ** $p \leq 0.01$; *** $p \leq 0.001$; **** $p \leq 0.0001$; ns not significant

cells with a combination of anti-CD20 and anti-CD19 antibodies abrogated the protective effect of CD4⁺ T cells in recipient A129 mice (Fig. 5c, d). Together, these results indicate that immune CD4⁺ T cell-induced protection requires type 2 IFN signaling, CD8⁺ T cells and B cells in the recipient mice.

**Neutralizing IgG protects against lethal ZIKV MR766 challenge.** To investigate the immune response capable to generate protective neutralizing antibodies, a series of serum and IgG transferences to naive mice and challenged with MR766 were performed (Fig. 6a). The transference of heat-inactivated serum obtained from A129 after 7 days of infection with ZIKV PE243,

but not from uninfected controls, to A129 recipient mice prevented weight loss and lethality induced by MR766 challenge (Supplementary Fig. 3a, b). Considering the high titers of anti-ZIKV IgG in these serums (Fig. 2d), we purified IgG, transferred to A129 mice and challenged with MR766. This treatment was also very effective in preventing weight loss and lethality (Fig. 6b, c). We next analyzed the role of IFNγ signaling in the generation of neutralizing antibodies to ZIKV. Passive immunization with serum from AG129 mice infected with PE243 was unable to protect A129 infected with MR766 (Fig. 6d, e). Given the essential role of CD4⁺ T on survival, the impact of CD4⁺ T cell depletion on the generation of ZIKV neutralizing antibodies upon PE243 infection was determined through in vivo and in vitro

heterologous neutralization assays. Compared to serum from PE243-infected mice, lack of CD4$^+$ T cells caused a major reduction in the neutralizing capacity of the serum, both in vivo as well as in vitro (Fig. 6f–h). These results paralleled the higher in vitro neutralizing capacity of serum from A129 mice infected with PE243 compared to serum from AG129-infected mice (Fig. 6h). Moreover, serum obtained after 7 days of ZIKV infection from AG129 mice or from mice depleted of CD4$^+$ T cells had significant lower titers of anti-ZIKV IgM, IgG and IgG2a isotype compared to serum from A129-infected mice (Supplementary Figs. 3c and 6i). Together, these results indicate an essential role of type 2 IFNR signaling and CD4$^+$ T cells in the resistance to ZIKV and suggest this effect is mediated in part by high serum titers of neutralizing anti-ZIKV IgG antibodies.

## Discussion

In the present study, we show that infection of 4-week-old type 1 IFNR-deficient mice with the ZIKV Brazilian strain, PE243, triggered an early and robust adaptive immune response, including the activation of a high percentage of CD4$^+$ T cells capable of producing IFNγ, increasing GC-B cells and anti-ZIKV neutralizing antibody titers, associated with reduction in viral titles in the CNS and survival. Notably, CD4$^+$ T cell and B cell responses and IFNγ signaling were fundamental to protection against primary ZIKV infection. AG129 mice or young A129 mice (3-week old) infected with PE243 presented weight loss, high lethality and viral titers in the brain, displaying a similar profile with the 4-week-old A129 mice infected with ZIKV MR766. These results suggest that the balance between viral neurotropism and an efficient immune response dictates the CNS viral load, tissue damage and disease outcome. Using adoptive cell transfer, we demonstrated a critical role of CD4$^+$ T cells from PE243-infected mice on heterologous resistance to a lethal ZIKV challenge, in a mechanism that required IFNγ signaling, CD8$^+$ T cells and B cells in the recipient mice. Importantly, serum from A129 infected with PE243, but not from AG129 or CD4$^+$ T cell-depleted mice, had high titers of anti-ZIKV IgG2a antibodies and protected against a lethal challenge with MR766.

A recent study demonstrated that a single point mutation in the prM protein of an Asian lineage of ZIKV arose before the 2013 outbreak in French Polynesia[49]. This S139N substitution, also present in the ZIKV PE243, has been maintained in the recent epidemics in American countries and correlates with higher infectivity of mouse and human neural progenitor cells, increased pathogenesis during mouse development, likely contributing to the increased incidence of congenital ZIKV syndrome in recent outbreaks. However, as we and others have shown, ZIKV strains from Asian lineage are less virulent to mice compared to MR766[19,22]. Analyzes of the overall amino acid composition of several ZIKV strains demonstrate maximum amino acid differences between MR766 and a strain isolated in the recent outbreak in the Americas[22]. ZIKV strain MR766 was generated after successive mouse passages, being therefore, adapted to this specie, what may impact on virus replication efficiency in different tissues and on higher virulence. Conversely, a ZIKV mutant with a deletion of ten nucleotides in the 3′ untranslated region of the genome is avirulent to A129 mice, generating a potent and long lasting neutralizing antibody and T cell-mediated responses[50]. Thus, the variable outcomes of different ZIKV strains in mice represent an interesting model to understand the mechanisms involved in protection and pathogenesis.

Previous studies using cell depletion and adoptive cell transfer approaches have demonstrated a protective role of CD8$^+$ T cells in mouse models of flavivirus infection, including DENV and ZIKV[26,38,40,41]. In addition to the demonstrated effect of CD8$^+$

T cells controlling ZIKV burden, a contribution of this cell population to the brain pathology of ZIKV-infected mice has been recently shown[51]. A recent study using IFNR1 sufficient mice primed through ZIKV infection in the periphery and secondary intracerebral challenge indicates that CD8$^+$ T cells are not essential to protection, although in conditions of low antibody levels effector CD8$^+$ T cells might contribute reducing viral loads[52]. PE243 infection caused an early activation of CD8$^+$ T cells, characterized by the expression of cytokines and effector cytotoxic molecules. Using cell depletion, we demonstrated the essential role of CD8$^+$ T cells in the primary response of A129 mice infected with PE243 and in the protective effect of adoptively transferred immune CD4$^+$ T cells upon the challenge with MR766. However, adoptive transfer of splenic CD8$^+$ T cells from A129 mice after 7 days of infection only modestly protects against the lethality caused by MR766.

We observed that depletion of CD4$^+$ T cells rendered A129 mice susceptible to ZIKV PE243, indicating an essential role of these cells in the response to the primary infection with ZIKV. Mechanistically, lack of CD4$^+$ T cells impaired the generation of neutralizing anti-ZIKV antibodies. Adoptive transfer of splenocytes obtained from mice infected for 1 week with PE243 conferred protection against a lethal challenge with ZIKV MR766. Purified immune B cells were not protective, while transfer of immune CD4$^+$ T cells fully protects recipient naïve A129 mice from M766 lethal infection, indicating a crucial role of CD4$^+$ T cells in the protective immune response to ZIKV. Of note, at 20 days postinfection, no viral RNA could be detected in the brains of mice that received purified immune CD4$^+$ T cells (data not shown). Our results also suggest that for ZIKV, compared to DENV, CD4$^+$ T cells are more relevant to the control of infection. Interestingly, depletion of CD4$^+$ T cells in C57BL/6 mice does not abrogate protective efficacy of gene-based vaccines for ZIKV[28]. A recent study demonstrated that despite full protection upon secondary intravaginal challenge, depletion of CD4$^+$ T cells reduces the generation of anti-ZIKV neutralizing antibodies[25]. Conversely, the absence of CD4$^+$ T cells in IFNR1 sufficient mice severely affects viral clearance of immunized mice, but not naive mice, upon an intracerebral challenge with ZIKV[52]. Future studies are required to define if these differences are associated with the vaccination/infection protocols or with the differences in the genetic backgrounds of mice used, including the role of IFNR1 signaling in adaptive immunity to ZIKV.

Our results point to an essential contribution of IFNγ signaling in the resistance to ZIKV infection. Previous studies also suggest an increase susceptibility to ZIKV of AG129 mice, but no characterization of the adaptive immune response were carried out. One study showed that AG129 mice are very susceptible to ZIKV infection, with viral dissemination to brain and testis, but no caparison to A129 has been performed[53]. Another study using AG129 in primary ZIKV infection was inconclusive determining the role of IFNγ in resistance[54]. The authors found no major differences in viral loads and lethality comparing A129 and AG129 mice, although a tendency of high severity of neurologic signs was noticed in AG129 mice. The lack of significant differences were likely due to the use of 3-week old mice, an age in which A129 are also very susceptible to ZIKV. In contrast to A129 mice, which control the infection with ZIKV PE243, we observed that AG129 mice were very susceptible, had high viral titers in the brain and were unable to generate an effective anti-ZIKV neutralizing antibody response. The susceptibility of AG129 to ZIKV PE243 paralleled the requirement of Type 1 and 2 IFN receptor signaling for the protection against DENV[43,55,56]. The protection provided by adoptively transferred immune splenocytes or purified CD4$^+$ T cells against the

challenge with ZIKV MR766 also required type 2 IFNR signaling in cells of the recipient mice. Interestingly, immunization of AG129 mice with a candidate vaccine of inactivated ZIKV conferred protection[57]. We speculate that this vaccine efficacy could be related to the immunization protocol that uses Alum as adjuvant and a regimen of prime and boost, thus bypassing the requirement of IFNγ. It is well known that IFNγ triggers several effector mechanisms that can contribute to viral clearance, including activation of cytotoxic CD8$^+$ T cells and Ig class switch recombination. Moreover, IFNγ might also affect the access of IgG to neuronal tissues and the consequent viral clearance. In mouse models of viral encephalitis by the neurotropic viruses herpes simplex virus type 2 and vesicular stomatitis virus, IFNγ produced by CD4$^+$ T cells is essential to open the BBB, allowing protective antiviral IgG antibodies to enter neuronal tissues[58]. We observed that susceptible AG129 mice had an increased amount of IgG in the brains at 5 days post-ZIKV infection[59], suggesting that, in this experimental model, IFNγ signaling is not essential to open the BBB.

The importance of B cells and neutralizing IgM antibodies generated independent of CD4$^+$ T cells in early resistance against a primary acute infection by WNV, has been elegantly shown by Diamond and colleagues[37,60]. However, the reduced amount of specific IgG response, observed in mice unable to secrete IgM ($sIgM^{-/-}$), might have contributed to the spread of WNV in the CNS and the tissue injury. In this model, WNV-specific IgG antibodies start to be detected at 8 days postinfection. We provided a series of evidences indicating that B cells are also fundamental to the adaptive immune response to ZIKV infection. Depletion of B cells demonstrated the contribution of these cells to the protection in primary ZIKV infection. Importantly, B cells were an essential component of the effector mechanism of protection mediated by adoptive transferred CD4$^+$ T cells to naïve mice upon heterologous ZIKV challenge. High titers of anti-ZIKV VLPs and anti-EDIII ZIKV IgG antibodies could be detected at 7 days post-PE243 infection and transfer of this serum or purified IgG protected against a lethal challenge with ZIKV MR766. These results suggest that in ZIKV PE243-infected mice CD4$^+$ T$_{FH}$ cells and anti-ZIKV B cells are retained in germinal centers, likely undergoing somatic hypermutation of their Ig variable regions, which eventually leads to affinity maturation. Although several studies demonstrated a protective role of neutralizing antibodies during ZIKV infection in passive and active immunization protocols, or due to cross-reactivity from previous DENV infection, the remarkable susceptibility of mice depleted of B cells to a nonlethal challenge with ZIKV PE243 unveiled the participation of antibodies in the early control of primary ZIKV infection. The fast IgG neutralizing antibody response in the first week of infection with the ZIKV strain PE243 was associated with reduced viral loads in the CNS and survival. Interestingly, rhesus monkeys also display high titers of ZIKV-specific binding and neutralizing antibodies after 7 days of infection[12].

In conclusion, our study demonstrate a critical role of CD4$^+$ T cells, type 2 IFN receptor signaling, CD8$^+$ T cells and B cells in the immune response and control of ZIKV infection in type 1 IFNR-deficient mice. A possible model to explain our collective results following PE243 infection is the one in which CD4$^+$ T$_{FH}$ cells induce the differentiation of B cells into GC-B cells in a mechanism dependent of IFNγ. This interaction further fosters affinity maturation and robust neutralizing antibody development in the germinal center.

## Methods

**Mice**. A129 mice, type 1 IFNR-deficient mice, and type 1 and type 2 IFNR-deficient mice (AG129) were used at 3–5 weeks old. All transgenic mice were on the 129/Sv background. Mice were bred and housed at mice facility of the Instituto de Microbiologia Paulo de Góes, Universidade Federal do Rio de Janeiro. The experimental procedures were carried out in accordance and approved by the Institutional Animal Care and Use Committee of the Centro de Ciências da Saúde of the (104/16; CEUA-UFRJ, Rio de Janeiro, Brazil). All efforts were made to minimize animals suffering. Mice of similar ages were randomized into control and treatment groups without any bias on parents, weight, size, or gender. Mice that died accidentally due to anesthesia were excluded from analyzes.

**Cell lines and viruses**. Vero cells (ATCC-CCL81; kindly given by Dr. Amilcar Tanuri, Instituto de Biologia, UFRJ) were cultured in DMEM (Life Technologies, Grand Island, NY) supplemented with L-glutamine (Sigma-Aldrich, St Louis, MO) and 5% fetal bovine serum (Life Technologies) and maintained at 37 °C with 5% CO$_2$. We used ZIKV strain PE243 (Brazil/South America, gene bank accession no. KX197192), isolated from a febrile case in the state of Pernambuco (2015), and ZIKV strain MR766 (Uganda/Africa, accession no. NC012532.1). The viruses were grown and titrated in Vero cells as previously described[61]. Vero cells were free of mycoplasma, as analyzed by PCR prior to use.

**Viral infection**. A129 and AG129 mice were randomized and anesthetized intraperitoneally using 100 mg/kg of ketamine 10% (syntec, SP, BR) and 10 mg/kg of xylazine 2% (Rhobifarma, SP, BR) prior to viral infection. Mouse infection was performed intravenously (retro-orbital plexus) using ZIKV strain MR766 or PE243 diluted in X1 phosphate-buffered saline (PBS), $2 \times 10^5$ plaque-forming units (PFU), 200 μL/animal. Uninfected mice were used as controls. Viral infection was determined by survival, weight loss and disease signs monitoring. Mice weighting less than 25% of their initial weight were euthanized to avoid unnecessary suffering. Survival curve data are shown as percentage of survival at the end of the experiment. Body weight gain is shown as percentage of difference between the weight in a given day and the initial weight of the same mouse. Viral RNA or infectious particles in the brain and spleen were detected by qRT-PCR and plaque assay, respectively. A group of A129 mice were infected simultaneously by intravenous route with $10^4$ PFU of ZIKV strain PE243 and $2 \times 10^5$ PFU of MR766 strain. The amount of ZIKV PE243 reproduces the same viral load observed in the mouse spleen 7 days postinfection with ZIKV PE243.

**Cell depletion**. B cells were depleted with a combination of anti-CD19 (Clone 1D3; BioXCell, West Lebanon, NH) and anti-CD20 (Clone 18B12; Biogen Idec, Weston, MA) inoculated intravenously (100 μL, 2.5 mg/kg) on A129 mice. A129 control mice were inoculated with IgG2a isotype control. CD4$^+$ T cells were depleted with anti-CD4 (Clone GK1.5; BioXCell) intraperitoneally (100 μL, 5.0 mg/kg) on A129 mice. CD8$^+$ T cells were depleted with anti-CD8 (Clone Lyt 2.1 BioXCell) intraperitoneally (100 μL, 5.0 mg/kg) on A129 mice. Viral challenge was performed 48 h after cell depletion.

**Cell isolation and enrichment**. ZIKV-infected mice were euthanized and splenocytes were homogenized followed by red blood cell lysis in 1 mL of ACK lysing buffer for 1 min. CD8$^+$ or CD4$^+$ T cells were isolated by negative or positive selection, respectively, using magnetic-beads, according to the manufacturer's instructions (MACS; Miltenyi biotech, Bergisch Gladbach, DE). B cells (B220$^+$CD138$^-$) were isolated from total splenocytes by fluorescence-activated cell sorting (FACS) technique on MoFlo XDP (Beckman Coulter Inc., Brea, CA). For detection of viral presence in lymphocytes obtained in the spleen, CD8$^+$ T cells (CD3$^+$CD8$^+$), CD4$^+$ T cells (CD3$^+$CD4$^+$) and B cells (B220$^+$CD138$^-$) were sorted on BD FACSAria II cytometer (BD Biosciences Immunocytometry Systems, San Jose, CA). Gate strategies for CD4$^+$, CD8$^+$ T and B cells are shown in Supplementary Fig. 4. Purity of B cells, CD4$^+$ and CD8$^+$ T cells ranged between 90 and 96%, as determined by postenrichment flow cytometry.

**Adoptive cell transfer**. A129 donor mice were infected with ZIKV PE243 as described above. At 7 days postinfection, splenocytes ($5 \times 10^7$/mouse), CD8$^+$ T cells ($1 \times 10^7$/mouse), CD4$^+$ T cells ($4–5 \times 10^7$/mouse) and B cells ($1 \times 10^7$/mouse) were isolated, and transferred in the indicated cell numbers. Naive splenocytes ($5 \times 10^7$/mouse) obtained from uninfected mice were used as controls during adoptive cell transfer experiments. Cells were transferred intravenously (retro-orbital plexus) into A129 or AG129 recipient mice, simultaneously to ZIKV MR766 challenge. CD8$^+$ T cell-depleted mice or B cell-depleted mice received CD4$^+$ T cell 48 h after depletion. Whenever indicated, ZIKV-immune splenocytes from donor mice were labeled using CFSE (Cell-Trace$^{TM}$ CFSE Cell Proliferation kit, Life Technologies)at a final concentration of 5 μM for 10 min at 37 °C, before adoptive cell transfer. Proliferation of donor B cells, CD4$^+$ and CD8$^+$ T cells on the spleen of recipient mice as well as IFNγ producing CD4$^+$ and CD8$^+$ T cells were determined by means of flow cytometry at 3 days postinfection.

**Passive sera administration**. AG129, A129, or A129 CD4[+] T cell-depleted donor mice were bled at 7 days postinfection with ZIKV strain PE243. The obtained sera were pooled and stored at −80 °C. Naïve serum obtained from uninfected A129 donor mice was used as control. Previous to transfer experiments, serum was heated at 56 °C for 30 min. For passive serum transfer experiments, ZIKV-immune serum or naïve serum (150–200 µL) was transferred to A129 recipient mice by intravenous route simultaneously to MR766 challenge.

**IgG purification and administration**. Sera from A129 mice were pooled and then purified in Protein G HP Spin TrapTM columns (GE Healthcare, Buckinghamshire, UK), following the manufacturer's instructions. Next, purified IgG was dialyzed against X1 PBS, with two changes every 4 h, followed by an overnight incubation. Purified IgG (100 uL, 35.7 mg/kg) was transferred to A129 recipient mice by intravenous route simultaneously to MR766 challenge.

**RNA isolation and qRT-PCR**. Viral load on mouse tissues (brain, spleen, gut, liver, and kidney) or splenocytes specific cell populations was assessed by qRT-PCR. RNA was isolated using TRIZOL reagent (Sigma-Aldrich), and the synthesis of cDNA was performed using high-capacity cDNA Archive Kit (Life Technologies), following the manufacturer's instructions. The cDNAs obtained were subjected to qRT-PCR using a StepOnePlus Real-time PCR system (Life Technologies) and Taqman Master Mix Reagents (Life Technologies). The primers and probe applied for viral load evaluation were specific for the E sequences, as follow: ZIKV 1086 5′-CCGCTGCCCAACACAAG-3′; ZIKV 1162c 5′-CCACTAACGTTCTTTTGCA-GACAT-3′; ZIKV 1107-FAM 5′-AGCCTACCTTGACAAGCAGTCAGA-CACTCAA-3′[62]. Standard curve was performed using cDNAs obtained from virus samples stocks, ranging from 75,000 to 0.75 PFU/mL, to estimate the genome copy number of ZIKV (RNA equivalent). Samples were deemed false if the cycle threshold cutoff value of 30 was exceeded. Interest genes expression was normalized using the housekeeping gene glyceraldehyde-3-phosphate dehydrogenase.

**Plaque assay**. Viral titers were determined by plaque assay in Vero cells. Briefly, the cells were cultured overnight at $4 \times 10^4$ cells/well in a 24-well plate. At approximately 70% of confluence, cells were incubated with tissue samples from ZIKV-infected mice in serial dilution, and cultured with 1.5% carboxy-methil-cellulose, in DMEM supplemented with 1% fetal calf serum (FCS). At the day 5 of culture, cells were fixed in formaldehyde 4%, followed by a washing step and staining with 1% crystal violet for 2 h. Plaques were counted and viral titers were calculated and expressed as PFU/mL.

**Neutralization assay**. Serum from ZIKV PE243 A129 or AG129-infected mice was collected periodically. For neutralization assays, ZIKV-infected mice sera were heated at 56 °C for 30 min. After sequential dilutions, the serum samples were incubated with ZIKV MR766 strain ($10^3$ PFU) at 37 °C for 2 h. Plaque assay was used as a detection system.

**Stimulation of splenic lymphocytes and flow cytometry**. Cells obtained from infected mice were harvested at $10^6$ cells/well in a 96-well plate. Splenocytes from uninfected mice were used as control. To assess intracellular cytokines and cytotoxic molecules, splenocytes were previously restimulated in vitro unspecifically or with viruses. For unspecific stimulation, cells were incubated with phorbol 12-myristate 13-acetate (PMA) (20 ng/mL) (Sigma-Aldrich) and ionomycin (4 µg/mL) (Merck Milipore, Billerica, MA) in the presence of Brefeldin A X1 (eBioscience, San Diego, CA) at 37 °C and 5% CO₂, during 4 h. In order to evaluate ZIKV-specific responses, cells from infected mice were harvested at $5 \times 10^5$ cells/well in a 96-well plate and restimulated in vitro using UV-inactivated virus and maintained in supplemented RPMI medium (Lonza, Baltimore, MD) at 37 °C with 5% CO₂ during 72 h. Brefeldin A X1 was added at the cultures on the last 8 h. After specific or unspecific in vitro stimulation, cells were stained with antibodies against surface markers conjugated with fluorochromes, during 30 min at 4 °C. Next, cells were washed and fixed in 1% paraformaldehyde (PFA). Finally, intracellular cytokines were stained with antibodies diluted in X1 permeabilization buffer (eBioscience) during 40 min at 4 °C. On experiments of cells immunophenotyping where in vitro stimulation was not necessary, permeabilized cells were stained with antibodies to evaluate transcriptional factors, while for some experiments, cells were stained only with antibodies against surface markers. Monoclonal antibodies used for cell surface and intracellular staining are shown on supplementary tables 1 and 2, respectively. To assess specific ZIKV CD8[+] T cells, splenocytes were stained with antibodies against surface markers and an anti-ZIKV protein E tetramer BV421-labeled at 37 °C for 1 h. The anti-ZIKV protein E tetramer, specifically against ZIKV E-4 (IGVSNRDFV) peptide, was synthesized via the NIH tetramer core facility. All Acquisitions were performed on BD FACSCanto II flow cytometer (BD Biosciences Immunocytometry Systems), and data were analyzed by using the FlowJo vX software (Tree Star Inc., Ashland, OR). Gate strategies for CD4[+] and CD8[+] T cells, B cells, plasma cells, NK cells, and anti-ZIKV E tetramer analyzes by flow cytometry are shown in Supplementary Figs. 5 and 6.

**Immunohistochemistry of brain tissues**. Animals were deeply anesthetized with xylazine/ketamine and then transcardially perfused with ice-cold 0.9% saline, followed by 4% PFA in PBS. Brains were rapidly removed from skulls, postfixed in PFA for 1 day at 4 °C, and cryoprotected in a PFA solution containing 20% (w/v) sucrose overnight. The frozen brains were then sectioned into 20 µm thick coronal sections using a sliding microtome (Leica Biosystems, Richmond, IL). Slices were collected in a cold cryoprotectant solution (0.05 M sodium phosphate buffer, pH 7.4, 30% ethylene glycol, 20% glycerol) and stored at −20 °C. For cerebral hemorrhages analyzes, free-floating brain sections were washed with 0.4% Triton X-100 in PBS (3 × 10 min) and then incubated with DAB (Vector Laboratories, Youngstown, OH) for 10 min. Tissues were thereafter washed with PBS (3 × 10 min), counterstained with thionin, cleared in HistoChoice® Clearing Agent (Sigma-Aldrich) and coverslipped with Organo/Limonene MountTM (Sigma-Aldrich). For IgG staining, free-floating sections were washed with 0.4% Triton X-100 in PBS (3 × 10 min) and then incubated for 30 min in a blocking solution containing 4% normal goat serum (Thermo Fisher Scientific,Waltham, MA) in PBS. Sections were washed with PBS (3 × 10 min), followed by a 2 h incubation with a biotinylated goat anti-mouse IgG (H + L) antibody (1:500; Vector Laboratories). Binding was visualized using the peroxidase-based Vectastain ABC kit and DAB (Vector Laboratories). Tissues were thereafter washed with PBS (3 × 10 min), cleared in HistoChoice® Clearing Agent (Sigma-Aldrich) and coverslipped with Organo/Limonene MountTM (Sigma-Aldrich). Slides were scanned with a Pannoramic MIDI II scanner (3DHISTECH Ltd., Budapest).

**Serum levels of anti-VLPs and anti-EDIII antibodies**. In the ELISA assays, VLPs mimicking the 3-D structure of the virus were used to coat the plates. These VLPs were produced at the Cell Culture Engineering Lab of COPPE/UFRJ by a stable pool of HEK293 cells constitutively expressing a prME construct designed according to the sequence of BeH819966 ZIKV strain (Genbank KU365779). VLPs were purified from serum-free cell culture supernatant by a two-step chromatography process prior to use. Briefly, ELISA plates were coated with 1 ug/mL of VLPs or 150 ng EDIII protein[30] in PBS per well and stored overnight at room temperature. Plates were then blocked with PBS 1% BSA for 2 h at room temperature. Serum samples were initially diluted 1:40 and serially threefold diluted to 1:1080 with PBS 1% BSA and added for 2 h at room temperature. Next, the following goat secondary antibodies conjugated with horseradish peroxidase were added to the plate for 2 h at room temperature: anti-mouse IgM (1:400), anti-mouse IgG (1:8000), anti-mouse IgG1 (1:2000) or anti-mouse IgG2a (1:2000) (SouthernBiotech, Birmingham, AL). Finally, TMB substrate solution (Life Technologies) were added and plates were read at 450 nm. Plates were washed with PBS each step.

**Statistical analysis**. Two-tailed Student's $t$ test was used to compute the significance between the groups. Survival experiments were analyzed using log rank test. All tests were performed on GraphPad Prism 6.00 (GraphPad Software, La Jolla, CA). (*) $p \le .05$; (**) $p \le .01$; (***) $p \le .001$; (****) $p \le .0001$; not significant (ns). Values for all measurements are expressed as mean or mean ± standard deviation. The sample size of animal studies was calculated by performing a power analysis using Piface software (Written by Russ Lenth, University of Iowa)[63]. Mouse experiments were performed with groups of 3−18 mice. Each experiment was usually repeated two or three times.

**Data availability**. All relevant data that support the findings of this study are available from the corresponding author upon request.

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

## Acknowledgments

We thank Dr. Amilcar Tanuri and Dr. Ernesto T.A. Marques Jr. for kindly donating the ZIKV strain MR766 and PE243, respectively. Zika EDIII recombinant protein was kindly provided by Dr. Michel Nussenzweig. We acknowledge Leda R. Castilho, Matheus O. Souza, Renata G. F. Alvim, and Tulio M. Lima from Cell Culture Engineering Lab at COPPE/UFRJ for providing the purified VLPs used in the ELISA assays. We thank the members of the Bozza lab for helpful discussions and suggestions. This work was supported by grants from CNPq, FAPERJ, CAPES, FINEP (M.T.B) and ICGEB (CRP/BRA16-05-EC; to R.M.P) C.G.O.L., J.Z.K., F.M.F., and V.G.S.S. are supported by fellowships from CAPES, M.P.P., and C.B.C. by CNPq and S.V.A.C. by FAPERJ.

## Author contributions

C.G.O.L. and J.Z.K. conducted the experiments, supervised and designed the experiments, interpreted experimental results, and wrote the manuscript. F.M.F. designed and conducted experiments, supervised, interpreted experimental results, and edited the manuscript. V.G.S.S., M.P.P., S.V.A.C., and C.B.C. conducted experiments, interpreted experimental results, and edited the manuscript. P.C.O., A.I., R.M.P., and L.B.A. designed experiments, supervised, interpreted experimental results, and edited the manuscript. H.A.P.N., P.M.P-C., and A.M.V. conducted the experiments, supervised and designed the

experiments, interpreted experimental results, and wrote the paper. M.T.B. supervised, designed, and interpreted experiments and wrote the manuscript.

## Additional information

**Competing interests:** The authors declare no competing interests.

