## [Peer Review File · Nature Communications]

Reviewers' comments:

Reviewer #1 (Remarks to the Author):

In this manuscript, the authors describe an important role for components of the adaptive immune response in controlling Zika virus (ZIKV) infection. The authors find that the neutralizing antibody response is dependent on a functional CD4 T cell compartment, and combined, aid in reducing viral titers in the brains and mediate protection. The authors performed adoptive transfers followed by heterologous protection studies within type I and type II interferon receptor knockout mice. While this study is important, there are several confounding concerns that limit the interpretation and relevance of the authors' findings. Further, the role of CD4 T cells have been well described for a number of flaviviruses, including WNV, JEV and DENV, and have included roles for mediating antibody responses, reducing virus replication, protection and mediating immunity (through regulatory T cells). Thus, the authors claim that a role for CD4+ T cells during flavivirus infection remains elusive is rather not accurate. Thus, the significance of these findings are dampened and it is rather obvious that CD4 T cells are important in mediating either antibody responses and/or controlling virus replication. The major concerns with this manuscript are listed below:

- 1) CD4+ T cells have been shown to be critical for a number of flaviviruses, including WNV, JEV and DENV. (first sentence in abstract).
- 2) The use of IFNAR mice to study ZIKV pathogenesis limits the interpretation of development of immunity to viral infections. It is difficult to reconcile the lack of type I IFN signaling during virus infection and the relevant induction of adaptive immunity. The authors should determine whether there is expanded cellular tropism for ZIKV in this model system (i.e. are B cells, CD4 T cells or CD8 T cells infected in the absence of type I or II IFN)?
- 3) Figure 1: While important, these findings are as expected, in that B cells, CD4 and CD8 T cells control ZIKV infection. In fact, several other groups have made similar observations using comparable mouse models of ZIKV infection. It is not clear how the authors evaluated intracellular IFN-gamma secretion in T cells. The authors should evaluate other relevant cytokines and effector molecules (e.g. TNF, IL-2, GrB, Perforin, etc...). Further, the authors should evaluate virus-specific cellular responses as the epitopes for CD8 T cells have been mapped and described in the literature. Lastly, the authors state that their model is non-lethal, however, there is approximately 10% death in this mouse model.
- 4) Figure 2: The role of IFN-g in controlling ZIKV infection is already well-described in the literature. IFN-gamma plays an important role at very early times during ZIKV infection. It is not clear how CD4 T cells would be critical during such an early time during infection. Further, the authors should perform a detailed virologic analysis rather than evaluating two tissue compartments on two timepoints. The authors infect mice through the IV route. Why was this route chosen over a subcutaneous route? Panel F) Are the T cell responses virus-specific?
- 5) Figure 3-4: While the authors perform a series of intriguing adoptive transfer studies, the analysis is greatly confounded by the potential of transferring virally infected cells into a naïve host. The authors should confirm that they are not transferring infected cells (splenocytes, T cells, B cells, DCs, Macs, etc...) into a naïve host. Further, it is not surprising that IFN-gamma and B cells are required for mediating protection against ZIKV infection.
- 6) Figure 5: The conclusion from the sera transfer studies are confounded as it is not clear whether the authors heat-inactivated the serum prior to transfer into a naïve host. Again, the role of antibodies is well-described in the literature for protecting against ZIKV infection. Thus, it is not clear what new information is gained through these studies. What are the proportions of virus-specific IgM and IgG subclass antibodies? How does this correspond to neutralizing antibodies? A greater depth of analysis here would potentially elevate these studies.

Reviewer #2 (Remarks to the Author):

Overall, this manuscript is interesting. However, some results are either over interpreted or need additional analysis or experiments for confirmation.

Major points:

1. Lack of statistical analysis: For example, the survival experiments in this study were not analyzed for statistical differences.
2. Many experiments were performed using very small group size (ex, n= 3 or 2 were used in Fig. 1, Fig. 2, & fig. 3)
3. For the adoptive transfer study shown in Fig. 3, the authors conclude that transfer of CD8+ T cells from ZIKV PE243- infected mice do not protect the recipient mice. In contrast, transfer of CD4+ T cells confers protection in the recipient mice. 4-5 x10⁷/mouse CD4+ T cells were used , which seems to be unusually high # for adoptive transfer. In comparison, they used ¼ to 1/5 of CD8+ T cells for transfer. The CD4/CD8 ratio is not equivalent to the physiological condition. Thus, the difference in cell # could also contribute to the survival rate differences in this experiment. In addition, CD8+ T cell transfer may delay and/or reduce the death in the recipient mice. However, this needs to be confirmed by statistical analysis.
4. Fig. 4D: Although it does not show full protection, CD4 T cell transfer in the B-cell depleted group may reduce or delay death. There is no statistical analysis in this experiment to test this possibility.
5. The amino acid differences between ZIKV MR766 and ZIKV PE243 strains are not discussed. The differences may contribute to the differential protective effects of T and B cells in the heterologous ZIKV infection studies.

Minor point:

Some cited references do not support their statement. For example, in the Introduction section, the authors stated that the cross-protective responses to ZIKV were observed by human antibodies to DENV. However, some cited studies showed DENV immune sera enhanced host susceptibility to ZIKV infection.

Point by point
NCOMMS-18-01544

Reviewer #1 (Remarks to the Author):

In this manuscript, the authors describe an important role for components of the adaptive immune response in controlling Zika virus (ZIKV) infection. The authors find that the neutralizing antibody response is dependent on a functional CD4 T cell compartment, and combined, aid in reducing viral titers in the brains and mediate protection. The authors performed adoptive transfers followed by heterologous protection studies within type I and type II interferon receptor knockout mice. While this study is important, there are several confounding concerns that limit the interpretation and relevance of the authors' findings. Further, the role of CD4 T cells have been well described for a number of flaviviruses, including WNV, JEV and DENV, and have included roles for mediating antibody responses, reducing virus replication, protection and mediating immunity (through regulatory T cells). Thus, the authors claim that a role for CD4+ T cells during flavivirus infection remains elusive is rather not accurate. Thus, the significance of these findings are dampened and it is rather obvious that CD4 T cells are important in mediating either antibody responses and/or controlling virus replication. The major concerns with this manuscript are listed below:

1) CD4+ T cells have been shown to be critical for a number of flaviviruses, including WNV, JEV and DENV. (first sentence in abstract).

We have changed the abstract accordingly.

2) The use of IFNAR mice to study ZIKV pathogenesis limits the interpretation of development of immunity to viral infections. It is difficult to reconcile the lack of type I IFN signaling during virus infection and the relevant induction of adaptive immunity. The authors should determine whether there is expanded cellular tropism for ZIKV in this model system (i.e. are B cells, CD4 T cells or CD8 T cells infected in the absence of type I or II IFN)?

Most experimental studies characterizing the adaptive immune response against ZIKV, including vaccine experiments, used *Ifnar1*^{-/-} mice or mice with mutations in the pathways upstream or downstream the type 1 IFN signaling (Lazear et al., 2016; Manangeeswaran et al., 2016; Tripathi et al., 2017; Dowall et al., 2017; Shan et al., 2017). Such approach has been largely used to characterize the adaptive immune response to DENV as well. Susceptibility of humans to ZIKV is in part due to the effect of ZIKV NS5 protein in increasing proteasome-mediated degradation of STAT2, a transcription factor essential to type 1 IFN receptor signaling (Grant et al., 2016, Kumar et al., 2016). Mouse STAT2 is not a target for ZIKV NS5 and thus immune competent mice are highly resistant to ZIKV infection. It would have been interestingly to use anti-IFN1R neutralizing antibodies in WT mice, however we could not afford the cost of these experiments. We have analyzed by

PCR the presence of ZIKV RNA in the purified populations used in the transfer experiments. We found that B cells (557 ZIKV RNA+ copies/10⁶ cells) and CD4⁺ T cells (258 ZIKV RNA+ copies/10⁶ cells) had significant levels of viral RNAs. As discussed in below (topic 5), we provided evidences that this viral contamination is not responsible for the protective effect of adoptive cell transfer populations.

3) Figure 1: While important, these findings are as expected, in that B cells, CD4 and CD8 T cells control ZIKV infection. In fact, several other groups have made similar observations using comparable mouse models of ZIKV infection. It is not clear how the authors evaluated intracellular IFN-gamma secretion in T cells. The authors should evaluate other relevant cytokines and effector molecules (e.g. TNF, IL-2, GrB, Perforin, etc...). Further, the authors should evaluate virus-specific cellular responses as the epitopes for CD8 T cells have been mapped and described in the literature. Lastly, the authors state that their model is non-lethal, however, there is approximately 10% death in this mouse model.

Although several studies demonstrated a protective role of neutralizing antibodies during ZIKV infection in passive and active immunization protocols, we are not aware of any published article demonstrating the remarkable susceptibility of mice depleted of B cells in the early control of primary ZIKV infection. In fact, the role of B cells in early WNV infection has been very well documented specially by Diamond's group. Moreover, concerning ZIKV infection the participation of CD4⁺ T cells in resistance has not been published and in theory could be non-essential as in DENV or multifaceted as in WNV, as discussed in the manuscript. We think that our revised manuscript provides a series of novel evidences demonstrating the role of CD4⁺ T cells in primary and secondary ZIKV infection.

As suggested, we detailed the protocol for intracellular IFN-gamma analysis in T cells and performed a series of new experiments to define cytokines and effector molecules expressed by CD8⁺ and CD4⁺ T cells (Figure 1). As suggested, we also performed a series of experiments with tetramer, with ZIKV and with VLPs to characterize the virus specific cellular and humoral responses. These experiments are shown in Figure 2. We think that these experiments made the manuscript findings more robust.

Indeed, in one experiment using the isotype control we observed 1 out of 10 mice dying upon infection with ZIKV 243. Thus, we changed the text accordingly.

4) Figure 2: The role of IFN-g in controlling ZIKV infection is already well-described in the literature. IFN-gamma plays an important role at very early times during ZIKV infection. It is not clear how CD4 T cells would be critical during such an early time during infection. Further, the authors should perform a detailed virologic analysis rather than evaluating two tissue compartments on two timepoints. The authors infect mice through the IV route. Why was this route chosen over a subcutaneous route? Panel F) Are the T cell responses virus-specific?

We are not aware of any published study that properly characterized the role of IFN γ in the immune response to ZIKV infection. Our results point to an essential contribution of IFN γ signaling in the resistance to ZIKV infection. Previous studies also suggest an increase susceptibility to ZIKV of AG129 mice, but no characterization of the immune response were carried out. One study showed that AG129 mice are very susceptible to ZIKV infection, with viral dissemination to brain and testis, but no comparison to A129 has been performed (Aliota et al., 2016b). Another study using AG129 in primary ZIKV infection was inconclusive determining the role of IFN γ in resistance (Rossi et al., 2016). The authors found no major differences in viral loads and lethality comparing A129 and AG129 mice, although a tendency of high severity of neurologic signs was noticed in AG129 mice. We speculated that lack of significant differences were due to the use of 3 week old mice, an age in which A129 are also very susceptible to ZIKV. Thus, we believe that our current study provides novel informations concerning the protective role of IFN-gamma signaling in primary infection (Figure 3C-G). We also showed that IFN-gamma signaling in recipient mice is essential to the protective effect of immune CD4 $^{+}$ T cells. The mechanism is associated with a role of IFN-gamma signaling on class switching and generation of neutralizing antibodies (Figure 6H, I; Supplementary Figure 3C).

As suggested, we performed a more detailed virologic analysis (Figure 1C; Figure 3E; Supplementary Figure 1C).

Previous studies on experimental mouse models of ZIKV infection used different routes of infection (ex. subcutaneous, intravenous and intraperitoneal), in some cases comparing the outcome, and found no major differences (Lazear et al., 2016; Elong Ngonu et al., 2017; Shan et al., 2017). We opted to use the i.v. route considering our technical skills to perform the experiments, thus obtaining the targeted effect with less animal/animal variation. More recently, we performed experiments infecting through the subcutaneous route and had similar results to the intravenous route in terms of weight loss and survival curves.

5) Figure 3-4: While the authors perform a series of intriguing adoptive transfer studies, the analysis is greatly confounded by the potential of transferring virally infected cells into a naïve host. The authors should confirm that they are not transferring infected cells (splenocytes, T cells, B cells, DCs, Macs, etc...) into a naïve host. Further, it is not surprising that IFN-gamma and B cells are required for mediating protection against ZIKV infection.

As suggested, we analyzed the presence of virus RNA in the purified populations and found ZIKV RNA in B cells and CD4 T cells, but not in CD8 T cells. As demonstrated in Supplementary Figure 2B and in Figure 4C, splenocytes and purified CD4 $^{+}$ T cells were

very efficient in protecting against the lethal challenge with the ZIKV MR766 strain. In order to define the role of PE243 present in the transfer procedure, we simultaneously infected A129 mice with PE243 and MR766. This coinfection was not protective, ruling out a major interference of PE243 contamination in the adoptive transfer protocols.

Although not completely surprising, we are not aware of published studies demonstrating the critical role of IFN-gamma signaling and of B cells in the resistance to primary ZIKV infection. The studies that performed similar experiments, although in different models, were discussed in the revised manuscript. We also demonstrated that in the absence of IFN-gamma signaling and of B cells, immune CD4+ T cells are no longer protective against a lethal challenge with ZIKV MR766 (Figure 5B, D). Finally, we performed similar experiments with recipient mice depleted of CD8+ T cells and showed the importance of these cells in the protective effect of transferred immune CD4+ T cells (Figure 5D).

6) Figure 5: The conclusion from the sera transfer studies are confounded as it is not clear whether the authors heat-inactivated the serum prior to transfer into a naïve host. Again, the role of antibodies is well-described in the literature for protecting against ZIKV infection. Thus, it is not clear what new information is gained through these studies. What are the proportions of virus-specific IgM and IgG subclass antibodies? How does this correspond to neutralizing antibodies? A greater depth of analysis here would potentially elevate these studies.

We now clearly stated that the serum used were heat-inactivated. Moreover, we performed an experiment using purified IgG obtained from mice at 7 days post infection with PE243. The purified IgG was also very effective in protecting against the challenge with MR766 (Figure 6C).

As suggested, we performed a series of experiments to define the proportions of virus-specific IgM and IgG subclass antibodies in PE243-infected A129, AG129 and anti-CD4 depleted mice (Figure 2D; Figure 6I; Supplementary Figure 3C). We demonstrated the essential role of IFN-gamma signaling and CD4+ T cells in class switching and neutralizing antibody generation.

We believe that the suggested experiments significantly increased the relevance of our findings.

Reviewer #2 (Remarks to the Author):

Overall, this manuscript is interesting. However, some results are either over interpreted or need additional analysis or experiments for confirmation.

Major points:

1. Lack of statistical analysis: For example, the survival experiments in this study were not analyzed for statistical differences.

We have performed the statistical analysis that were lacking and included in the manuscript.

2. Many experiments were performed using very small group size (ex, n= 3 or 2 were used in Fig. 1, Fig. 2, & fig. 3)

As suggested, we have repeated the experiments to have larger groups.

3. For the adoptive transfer study shown in Fig. 3, the authors conclude that transfer of CD8+ T cells from ZIKV PE243- infected mice do not protect the recipient mice. In contrast, transfer of CD4+ T cells confers protection in the recipient mice. $4-5 \times 10^7$ /mouse CD4+ T cells were used, which seems to be unusually high # for adoptive transfer. In comparison, they used $\frac{1}{4}$ to $\frac{1}{5}$ of CD8+ T cells for transfer. The CD4/CD8 ratio is not equivalent to the physiological condition. Thus, the difference in cell # could also contribute to the survival rate differences in this experiment. In addition, CD8+ T cell transfer may delay and/or reduce the death in the recipient mice. However, this needs to be confirmed by statistical analysis.

Indeed, the proportions of CD4/CD8 used to the adoptive transfer experiments were not physiological. We agree that increasing the number of transferred cells could substantially affect the results. However, we defined the numbers of transferred cell populations based in the cell numbers found in the spleens of mice at 7 days post infection with the ZIKV PE243. The proportion found in the spleens of A129 mice after 7 days of infection with ZIKV was 4 CD4 to 1CD8.

We have analyzed the effect of CD8 T cell transfer and in fact this transfer significantly protected, albeit modestly, against the challenge with MR766.

4. Fig. 4D: Although it does not show full protection, CD4 T cell transfer in the B-cell depleted group may reduce or delay death. There is no statistical analysis in this experiment to test this possibility.

As suggested, we performed all statistical analysis including the lethality curves. Moreover, we characterized the effect of CD8+ T cell depletion in the protective effect of CD4+ T cell adoptive transfer experiments (Figure 5D).

5. The amino acid differences between ZIKV MR766 and ZIKV PE243 strains are not discussed. The differences may contribute to the differential protective effects of T and B cells in the heterologous ZIKV infection studies.

Indeed, the sequence differences of the Asian isolates including PE243 and the African isolate MR766 are very extensive, contributing to the virulence, tissue tropism and

pathology. It is also very important to consider the contribution of these sequence differences in immune evasion. We included the points in the discussion.

Minor point:

Some cited references do not support their statement. For example, in the Introduction section, the authors stated that the cross-protective responses to ZIKV were observed by human antibodies to DENV. However, some cited studies showed DENV immune sera enhanced host susceptibility to ZIKV infection.

As suggested, we included the findings regarding ADE from Stettler et al., 2016 and Bardina et al 2017 in the introduction.

Rio de Janeiro, May 08, 2018

Marcelo T. Bozza

REVIEWERS' COMMENTS:

Reviewer #1 (Remarks to the Author):

The authors have addressed the previous reviewers concerns.